# Evaluation of the Inhibitory Effects of (E)-1-(2-hydroxy-4,6-dimethoxyphenyl)-3-(naphthalen-1-yl)prop-2-en-1-one (DiNap), a Natural Product Analog, on the Replication of Type 2 PRRSV In Vitro and In Vivo

**DOI:** 10.3390/molecules24050887

**Published:** 2019-03-03

**Authors:** Amina Khatun, Sun You Park, Nadeem Shabir, Salik Nazki, A-Rum Kang, Chang-Gi Jeong, Byoung-Joo Seo, Myeon-Sik Yang, Bumseok Kim, Young Ho Seo, Won-Il Kim

**Affiliations:** 1College of Veterinary Medicine, Chonbuk National University, Iksan 54596, Korea; aminak@jbnu.ac.kr (A.K.); saliknazki@jbnu.ac.kr (S.N.); kar0913@jbnu.ac.kr (A.-R.K.); jcg0102@gmail.com (C.-G.J.); byoungjooseo@gmail.com (B.-J.S.); 111@jbnu.ac.kr (M.-S.Y.); bskims@jbnu.ac.kr (B.K.); 2Department of Pathology, Faculty of Animal Science and Veterinary Medicine, Sher-e-Bangla Agricultural University, Dhaka 1207, Bangladesh; 3College of Pharmacy, Keimyung University, Daegu 42601, Korea; oasis4488@naver.com; 4Division of Animal Biotechnology, Faculty of Veterinary Sciences and Animal Husbandry, Sher-e-Kashmir University of Agricultural Sciences and Technology of Kashmir, Srinagar 190025, India; nadeemshabir@jbnu.ac.kr

**Keywords:** antiviral therapeutics, DiNap, PRRSV, VR2332, pigs

## Abstract

DiNap [(*E*)-1-(2-hydroxy-4,6-dimethoxyphenyl)-3-(naphthalen-1-yl)prop-2-en-1-one], an analog of a natural product (the chalcone flavokawain), was synthesized and characterized in this study. Porcine reproductive and respiratory syndrome virus (PRRSV) is the most challenging threat to the swine industry worldwide. Currently, commercially available vaccines are ineffective for controlling porcine reproductive and respiratory syndrome (PRRS) in pigs. Therefore, a pharmacological intervention may represent an alternative control measure for PRRSV infection. Hence, the present study evaluated the effects of DiNap on the replication of VR2332 (a prototype strain of type 2 PRRSV). Initially, in vitro antiviral assays against VR2332 were performed in MARC-145 cells and porcine alveolar macrophages (PAMs). Following this, a pilot study was conducted in a pig model to demonstrate the effects of DiNap following VR2332 infection. DiNap inhibited VR2332 replication in both cell lines in a dose-dependent manner, and viral growth was completely suppressed at concentrations ≥0.06 mM, without significant cytotoxicity. Consistent with these findings, in the pig study, DiNap also reduced viral loads in the serum and lungs and enhanced the weight gain of pigs following VR2332 infection, as indicated by comparison of the DiNap-treated groups to the untreated control (NC) group. In addition, DiNap-treated pigs had fewer gross and microscopic lesions in their lungs than NC pigs. Notably, virus transmission was also delayed by approximately 1 week in uninfected contact pigs within the same group after treatment with DiNap. Taken together, these results suggest that DiNap has potential anti-PRRSV activity and could be useful as a prophylactic or post-exposure treatment drug to control PRRSV infection in pigs.

## 1. Introduction

Porcine reproductive and respiratory syndrome (PRRS) is the most economically important infectious disease for the swine industry worldwide. PRRS is characterized by reproductive failure in sows and respiratory disease in pigs of all ages [1,2]. The annual loss associated with PRRS is approximately 664 million USD, as estimated by swine producers in the United States [3]. PRRS virus (PRRSV) is a member of the *Arteriviridae* virus family in the order *Nidovirales*, along with equine arteritis virus (EAV), murine lactate dehydrogenase-elevating virus (LDV), and simian hemorrhagic fever virus (SHFV) [4,5]. PRRSV is a small, enveloped virus with a single-stranded, nonsegmented, positive-sense RNA genome that is approximately 15 kb in length [5,6]. The genome of PRRSV encodes at least ten open reading frames (ORFs) called ORF1a, ORF1b, ORF2a, ORF2b, ORF3, ORF4, ORF5a, ORF5, ORF6, and ORF7 [7,8]. PRRSVs used to be grouped into European (type 1) and North American (type 2) genotypes, but currently classified as two species, betaarterivirus suid 1 and betaarterivirus suid 2, respectively. There is substantial genetic and antigenic variation among PRRSV strains, even within the same virus species [9,10]. Since the first emergence of PRRS more than two decades ago, numerous efforts have been made to develop an effective control measure for this challenging disease. Currently, vaccination is the primary control method, and is widely used to control PRRS in pigs. However, the use of commercial vaccines with either modified live virus (MLV) or killed virus is inadequate to control PRRSV infection [11] because of the highly divergent PRRSV strains in the field. Antiviral therapeutics could be an alternative or additional control measure to combat these viral infections when a successful vaccine that matches the circulating virus is not available. Therefore, pharmacological intervention might be a suitable and useful control strategy for PRRS. Many previous studies reported that a variety of natural compounds had been proven to exert antiviral activity against a number of viruses, such as equine herpesvirus 1 (EHV-1) [12], hepatitis B virus [13], human cytomegalovirus (HCMV) [14], and respiratory infections including the common cold [15] and PRRSV [16,17,18,19,20,21,22,23,24,25,26,27,28,29]. However, no such effective drugs are commercially available to prevent PRRSV infection. Chalcones (1,3-diaryl-2-propen-1-ones), which constitute a key class of natural products belonging to the *Flavonoid* family [30,31,32,33], were previously reported to have a wide range of biological activities, such as antiviral, antibacterial, antifungal, immunomodulatory, anti-inflammatory, antitumor, insect antifeedant, and antimutagenic activities [31,32,33,34,35,36,37,38]. Thus, the present study aimed to evaluate the antiviral effects of DiNap [(*E*)-1-(2-hydroxy-4,6-dimethoxyphenyl)-3-(naphthalen-1-yl)prop-2-en-1-one], a synthetic analog of the chalcone flavokawain, on PRRSV replication in cell culture systems. A study was then carried out in a pig model to demonstrate the inhibitory effects of DiNap on PRRSV infection.

## 2. Results

### 2.1. Chemistry

The synthesis of the flavokawain analog DiNap is presented in Scheme 1, which began with the preparation 2-hydroxy-4,6-dimethoxyacetophenone (**2**). Compound **2** was quantitatively obtained from the reaction of 2,4,6-trihydroacetophenone (**1**) with dimethyl sulfate in the presence of potassium carbonate in acetone. The corresponding acetophenone **2** was then subjected to the Claisen–Schmidt condensation reaction with 1-naphthaldehyde (**3**) in the presence of potassium hydroxide in methanol to provide flavokawain analogs (DiNap) with a yield of 32%.

### 2.2. In Vitro Evaluation of the Effects of DiNap on PRRSV Replication

The antiviral effect of DiNap on PRRSV replication was evaluated in MARC-145 cells and porcine alveolar macrophages (PAMs). DiNap reduced VR2332 replication in both cell lines in a dose-dependent manner (Figure 1). In MARC-145 cells, VR2332 replication was decreased by approximately 10- or 1000-fold in the presence of DiNap at concentrations of 0.02 or 0.04 mM, respectively. Viral growth was completely suppressed at a 0.06 mM concentration of DiNap, while the lowest concentration (0.01 mM) did not exert a significant effect on viral replication (Figure 1A). In addition, DiNap did not show significant toxicity in MARC-145 cells, even at a concentration of 0.06 mM at up to 48 h post treatment (hpt) (Figure 1C). Consistent with the results obtained in MARC-145 cells, in PAMs, DiNap also reduced VR2332 replication by approximately 10- or 100-fold in the presence of the 0.02 or 0.04 mM concentrations; the highest concentration (0.06 mM) caused the complete suppression of VR2332 replication, while 0.01 mM DiNap was not effective (Figure 1B). Similarly, DiNap did not cause significant toxicity in PAMs, even at concentrations of 0.06 mM at up to 48 hpt (Figure 1D).

### 2.3. Inhibitory Effect of DiNap on PRRSV in Experimentally Infected Pigs

#### 2.3.1. Serum Viremia

The viral loads in the serum samples were quantified by real-time reverse transcription-polymerase chain reaction (qRT-PCR) (TaqMan^®^), and the results are summarized in Figure 2. One pig (1, 4 or 7) in each group was challenged with VR2332 and maintained in the group with the other two unchallenged pigs to observe the effects of DiNap on virus transmission between challenged and unchallenged pigs.

In group 1 (NC), the pig (1) challenged with VR2332 had the highest viral load, with virus titers of 10^5.55^ TCID_50_/mL at 7 days post challenge (dpc), which gradually decreased to 10^1.39^ TCID_50_/mL at 28 dpc. Similarly, the two unchallenged pigs (2 and 3) in the same group also showed peak viral titers of 10^3.83^ and 10^4.64^ TCID_50_/mL at 7 dpc, which also gradually decreased to 10^1.23^ and 10^2.01^ TCID_50_/mL, respectively, at 28 dpc (Figure 2A).

In group 2 (DiNap at 0.04 mM/kg body weight), the pig (4) challenged with VR2332 had a peak viral titer of 10^4.18^ TCID_50_/mL at 7 dpc, which was more than 10-fold lower than the titers found in the pig (1) challenged with the virus in group 1; the viral titer gradually decreased to 10^0.93^ TCID_50_/mL at 28 dpc. Among the two unchallenged pigs in the same group, one pig (5) showed a peak virus titer of 10^3.25^ TCID_50_/mL at 7 dpc, which was also lower than the titers in both unchallenged pigs (2 and 3) in group 1, and the titer gradually declined to 10^1.39^ TCID_50_/mL at 28 dpc. The other unchallenged pig (6) did not show viremia at up to 7 dpc, although the virus was detected at 14 dpc, with a titer of 10^3.14^ TCID_50_/mL, which slowly decreased to 10^2.19^ TCID_50_/mL at 28 dpc (Figure 2A).

Consistent with these results, in group 3 (DiNap at 0.2 mM/kg body weight), the pig (7) challenged with VR2332 had peak viremia with a virus titer of 10^4.08^ TCID_50_/mL at 7 dpc, which was more than 10-fold lower than the titer found in the pig (1) challenged with virus in group 1, and the virus titer gradually decreased to 10^1.02^ TCID_50_/mL at 28 dpc. Neither unchallenged pig (8 and 9) in the same group displayed viremia at up to 7 dpc, although the virus was detected in both of the pigs at 14 dpc, with titers of 10^2.17^ and 10^2.92^ TCID_50_/mL, which slowly declined to 10^1.24^ and 10^1.51^ TCID_50_/mL, respectively, at 28 dpc (Figure 2A).

In addition, the average virus titers were measured in the pigs in each group. The pigs in group 1 (NC) exhibited an average peak viremia titer of 10^4.67^ TCID_50_/mL at 7 dpc, which gradually decreased to 10^1.54^ TCID_50_/mL at 28 dpc. In contrast, the pigs in group 2 (DiNap at 0.04 mM) and 3 (DiNap at 0.2 mM) had average titers of 10^2.47^ TCID_50_/mL and 10^1.36^ TCID_50_/mL, respectively, at 7 dpc, which were approximately 100-fold (*p* < 0.05) and 1000-fold (*p* < 0.001) lower than the average titer observed in the pigs in group 1 (NC) (Figure 2B). These results suggest that DiNap has significant antiviral effects against PRRSV, decreasing viral transmission and replication in pigs.

#### 2.3.2. Anti-PRRSV IgG Response

The PRRSV-specific antibody (IgG) response was measured in serum samples by enzyme-linked immunosorbent assay (ELISA) based on the nucleocapsid (N) protein. As summarized in Figure 3, all pigs in each group were seronegative at up to 7 dpc. Following infection, all 3 pigs (1 to 3) in group 1 (NC) became seropositive at 14 dpc, and the S/P (sample-to-positive) ratio gradually increased up to 28 dpc. Similarly, in group 2, two pigs [virus challenged (4) and unchallenged (5)] became seropositive at 14 dpc. The other unchallenged pig (6) in the same group was seronegative at up to 14 dpc and became seropositive at 21 dpc. In group 3, the challenged pig (7) became seropositive at 14 dpc, similar to the challenged pig (1) in group 1. However, the two other unchallenged pigs (8 and 9) in the same group were seronegative at up to 14 dpc and became seropositive at 21 dpc. There were no differences in the antibody (IgG) titers found among the groups (Figure 3). These results suggested that DiNap delayed viral transmission among pigs infected with PRRSV.

#### 2.3.3. Average Daily Weight Gain

The effect of DiNap on the growth rate of the pigs was evaluated over the course of PRRSV infection. The average daily weight gain (ADWG) was measured for all pigs in each group up to 28 dpc. DiNap significantly enhanced the ADWG for both treatment groups (groups 2 and 3) compared with that of the pigs in group 1 (NC) (Figure 4).

#### 2.3.4. Lung Pathology and Residual Viral Loads in Lung Tissues

All pigs in the untreated control group (group 1) exhibited higher lung lesion scores (gross and microscopic) than those in group 2 and group 3 (Figure 5).

The residual viral loads were also measured in the lung tissues collected from the pigs in each group. DiNap appeared to reduce the viral loads observed in the lung tissues of the pigs in each treatment group (groups 2 and 3) compared to those in group 1 (NC). The pigs in groups 2 and 3 exhibited average virus titers of 10^3.28^ TCID_50_/mL and 10^2.68^ TCID_50_/mL, respectively, which were approximately 10- and 100-fold lower than the titer of 10^4.65^ TCID_50_/mL observed in group 1. In group 1, the pig (1) challenged with VR2332 had a virus titer of 10^4.84^ TCID_50_/mL in its lung tissue, and the unchallenged pigs (2 and 3) in the same group exhibited virus titers of 10^4.94^ and 10^4.17^ TCID_50_/mL, respectively. However, in group 2, the pig (4) challenged with the virus had a virus titer of 10^2.10^ TCID_50_/mL, which was approximately 100-fold lower than the titer found in the challenged pig (1) in group 1. The two unchallenged pigs (5 and 6) in the same group had titers of 10^3.51^ and 10^4.24^ TCID_50_/mL, respectively, which were also lower than or similar to the titers found in both unchallenged pigs (2 and 3) in group 1. In group 3, the pig (7) that was challenged with the virus exhibited a virus titer of 10^1.98^ TCID_50_/mL in its lung tissues, which was also more than 100-fold lower than the titer found in the pig (1) that was challenged by the virus in group 1. Moreover, both unchallenged pigs (8 and 9) in the same group also had lower virus titers of 10^2.98^ and 10^3.11^ TCID_50_/mL, respectively, in their lung tissues than the titers found in both unchallenged pigs (2 and 3) in group 1 (Figure 6).

## 3. Discussion

The chalcone flavokawain was found to exhibit promising anticancer and anti-inflammatory properties in previous studies [39,40]. DiNap, a synthetic analog of the chalcone flavokawain, was evaluated in this study for its potential antiviral activity against PRRSV infection in cell culture systems and in a pig model. Chemically, DiNap is a synthetic analog of flavokawain, containing a chalcone scaffold of 1,3-diaryl-2-propen-1-one. DiNap has a 2-hydroxy-4,6-dimethoxybenzene moiety at the C1 position of 2-propene-1-one, which is a common structural feature of the naturally occurring compounds flavokawain A, B, and C. DiNap also contains a naphthalenyl moiety at the C3 position of 2-propene-1-one, which has been found in several antiviral agents [41,42]. In addition, chalcones, the main pharmacophore of DiNap, have been reported to show a wide spectrum of biological activities, including antiviral [43], antioxidative [44], and anti-inflammatory activities [45]. These previous studies are further supported by the results presented in the current study. DiNap effectively suppressed VR2332 replication in MARC-145 cells and PAMs at concentrations ranging from 0.02 to 0.06 mM, without showing significant cytotoxicity, even at up to 48 hpt. It is worth noting that the highest concentration of DiNap (0.06 mM) used in this study completely suppressed VR2332 replication in both cell lines (Figure 1). Although approximately 20% cell death was observed when DiNap was added at 0.06 mM, no significant side effects or toxicity were observed after treatment in pigs.

In the pig experiment, we demonstrated that DiNap exerted its potential inhibitory effects against PRRSV infectivity in a dose-dependent manner. Compared to the pigs in the untreated control group (group 1), the pigs treated with DiNap in group 2 and group 3 had reduced viral loads in the serum and lungs and increased weight gain (Figure 2, Figure 4 and Figure 6). Consistent with these findings, the pigs treated with DiNap also had fewer lesions (gross and microscopic) in their lungs than the control pigs (Figure 5), suggesting that DiNap exerted significant effects on PRRSV infection. More importantly, two injections of DiNap were able to delay virus transmission by approximately 1 week with direct contact between the infected (virus challenged) and uninfected (not virus challenged) pigs in both treatment groups (groups 2 and 3). However, the virus was easily transmitted between the infected and uninfected pigs in the untreated control group (group 1), indicating that DiNap could be useful as a prophylactic or postexposure treatment agent in cases of PRRSV infection. Because DiNap was produced in limited quantities on a laboratory scale, the antiviral effect of DiNap was only preliminarily evaluated with a small number of pigs in the current study. Therefore, more detailed animal experiments are needed to determine how long the antiviral effect of DiNap lasts in the future.

Taken together, these results demonstrated that DiNap might be useful as a potential prophylactic and postexposure anti-PRRSV drug for the treatment of PRRSV-infected pigs, although the efficacy and safety of long-term use of higher concentrations of DiNap remains to be confirmed in future experiments.

## 4. Materials and Methods

### 4.1. Synthesis of DiNap, an Analog of a Natural Product

#### 4.1.1. General Methods

All reagents and solvents were purchased from Sigma-Aldrich (Milwaukee, WI, USA) and Alfa Aesar (Ward Hill, MA, USA). All experiments involving moisture-sensitive compounds were carried out in an argon atmosphere. Analytical thin-layer chromatography was performed on precoated silica gel F_254_ thin layer chromatography (TLC) plates (E, Merck, Kenilworth, NJ, USA) with visualization under UV light. The final products were purified by preparative medium-pressure liquid chromatography (MPLC) (Biotage Isolera One instrument) on a silica column (Biotage SNAP HP-Sil) spectrometer (Biotage, Uppsala, Sweden). Nuclear magnetic resonance (NMR) analyses were performed using a Bruker 400 (400 MHz for ^1^H; 100 MHz for ^13^C) spectrometer (Bruker, Rheinstetten, Germany). The chemical shifts are reported in parts per million (δ). The deuterium lock signal of the sample solvent was used as a reference, and the coupling constants (*J*) are given in hertz (Hz). The splitting pattern abbreviations are as follows: s, singlet; d, doublet; t, triplet; q, quartet; dd, doublet of doublets; and m, multiplet. The purity of all tested compounds was confirmed to be higher than 95% by analytical high-performance liquid chromatography (HPLC) analysis performed with a dual-pump Shimadzu LC-6AD system equipped with a VP-ODS C18 column (4.6 mm × 250 mm, 5 μm, Shimadzu).

#### 4.1.2. 1-(2-Hydroxy-4,6-dimethoxyphenyl)ethanone (**2**)

A mixture of 2′,4′,6′-trihydroacetophenone hydrate (2.0 g, 11.89 mmol), potassium carbonate (3.62 g, 26.17 mmol) and dimethylsulfate (2.32 mL, 24.38 mmol) in acetone (30 mL) was stirred under argon at 66 °C for 2 h. The resulting mixture was cooled to room temperature, filtered, and washed with acetone. The filtrate was extracted with ethyl acetate. The organic layer was washed three times with saturated NaHCO_3_ solution, dried over Na_2_SO_4_, and concentrated under reduced pressure to produce compound **2** with 100% yield (R_f_ = 0.34 (1:9 ethyl acetate: hexane); ^1^H NMR (400 MHz, CDCl_3_) δ 14.06 (s, 1H), 6.05 (d, *J* = 2.0 Hz, 1H), 5.81 (d, *J* = 2.4 Hz, 1H), 3.85 (s, 3H), 3.81 (s, 3H), 2.61 (s, 3H)).

#### 4.1.3. (*E*)-1-(2-Hydroxy-4,6-dimethoxyphenyl)-3-(naphthalen-1-yl)prop-2-en-1-one (DiNap)

DiNap was synthesized following the procedure reported in the literature with slight modifications [46]. Briefly, a mixture of compound **2** (0.2 g, 1.02 mmol), 1-naphthaldehyde (0.14 mL, 1.02 mmol) and potassium hydroxide (1 g) in methanol (20 mL) was stirred at 50 °C for 3 days. The mixture was extracted with ethyl acetate. The organic layer was washed three times with water, dried over Na_2_SO_4_, concentrated under reduced pressure, and purified by MPLC to produce DiNap with 32% yield (R_f_ = 0.27 (1:9 ethyl acetate: hexane); ^1^H NMR (400 MHz, CDCl_3_) δ 14.40 (s, 1H), 8.61 (d, *J* = 15.6 Hz, 1H), 8.31 (d, *J* = 8.4 Hz, 1H), 7.96 (d, *J* = 15.2 Hz, 1H), 7.90–7.87 (m, 2H), 7.83 (d, *J* = 7.2 Hz, 1H), 7.60–7.49 (m, 3H), 6.13 (d, *J* = 2.0, 1H), 5.96 (d, *J* = 2.4 Hz, 1H), 3.89 (s, 3H), 3.823 (s, 3H); ^13^C NMR (100 MHz, CDCl_3_) δ 192.8, 168.7, 166.6, 162.8, 139.3, 134.0, 133.2, 132.0, 130.6, 130.4, 129.0, 127.0, 126.4, 125.7, 125.4, 123.9, 106.6, 94.1, 91.5, 57.4, 56.1).

After synthesis, DiNap was dissolved in dimethyl sulfoxide (DMSO) (Sigma, Aldrich, St. Louis, MO, USA) at a stock concentration of 20 mM, sterile-filtered using a 0.20-µm syringe filter (Corning, Germany), aliquoted and stored at 4 °C until use.

### 4.2. Viruses and Cells

VR2332, a prototype strain of type 2 PRRSV, was used in the present study. MARC-145, an African Green Monkey Kidney cell line (highly permissive to PRRSV) [47], and PAMs (the primary target cell for PRRSV replication in pigs) [48,49] were used for the in vitro experiments. PAMs were collected from 6-week-old, PRRSV-free pigs by bronchoalveolar lavage, as described previously [50,51]. MARC-145 cells were used for virus propagation, while the antiviral assays were performed in both cell lines (MARC-145 and PAMs). Both cell lines were maintained in RPMI-1640 medium (Gibco, Invitrogen, Carlsbad, CA, USA) supplemented with heat-inactivated 10% fetal bovine serum (FBS, Gibco, Invitrogen), 2 mM L-glutamine, and 100x antibiotic-antimycotic solution (Anti-anti, Invitrogen; 1x solution contains 100 IU/mL penicillin, 100 µg/mL streptomycin, and 0.25 µg/mL Fungizone^®^ (amphotericin B)) (named “growth medium”) at 37 °C in a humidified 5% CO_2_ atmosphere.

### 4.3. Antiviral Effect of DiNap Against PRRSV Replication in Cells

MARC-145 cells and PAMs were prepared in 6-well cell culture plates (BD, Falcon) 24 to 48 h before the start of the experiment. Following this, the cell monolayers were pretreated for 2 h before infection with growth medium containing DiNap at five different concentrations (0, 0.01, 0.02, 0.04, and 0.06 mM). After the pretreatment incubation, the cell monolayers were washed with growth medium, inoculated with VR2332 at a multiplicity of infection (MOI) of 0.01 and incubated for 1 h. After the final 1-h incubation, the viral inoculum was discarded, and the cell monolayers were replenished with growth medium containing the same concentrations of DiNap as a treatment and incubated for 4 more days under the same culture conditions. The cell culture supernatants were collected every 24 h, centrifuged, and stored at −80 °C until analysis. The progeny virus titers were measured using a microtitration infectivity assay [52]; the virus titration assay was described briefly in a previous study [53]. The virus titers were calculated based on the cytopathic effect (CPE) and are expressed as TCID_50_/mL [54].

### 4.4. MTT Assay

The cytotoxic effect of DiNap was assessed in MARC-145 cells and PAMs using an MTT [3(4,5-dimethylthiazol-2-yl)-2,5-diphenyltetrazolium bromide] (MTT, Sigma-Aldrich, St. Louis, MO, USA) assay, as described previously [55,56]. In brief, MARC-145 cells and PAMs were prepared in 96-well cell culture plates (BD, Falcon) and treated with growth medium containing DiNap (0, 0.01, 0.02, 0.04, and 0.06 mM) for 24 and 48 h. After incubation, the cell supernatants were discarded, and 20 µL of freshly made MTT (5 mg/mL) solution was added along with growth medium to each well. Then, the cells were incubated for another 3 h under the same conditions. After incubation, the MTT solutions were removed, and the plates were dried at 65 °C overnight. Then, 200 µL of dimethyl sulfoxide (DMSO; Hybri-Max^®^, Sigma-Aldrich) was added to each well to dissolve the formazan crystals. The plates were further incubated at 37 °C for 5 to 10 min to dissolve any air bubbles before the MTT signal was measured using an ELISA microplate reader at an absorbance of 540 nm.

### 4.5. Animal Experiment and Samples

In total, 9 three-week-old PRRSV-negative pigs were purchased. On their arrival at the facility, the pigs were randomly housed in three groups (3 pigs per group), designated group 1 (pigs 1 to 3), group 2 (pigs 4 to 6), and group 3 (pigs 7 to 9) and allowed to acclimatize for 3 days. Then, qRT-PCR (TaqMan) and ELISA were used to confirm that the pigs were negative for PRRSV. On day 0, one pig (1, 4, and 7 in groups 1, 2,and 3, respectively) in each group was challenged with VR2332 at a titer of 10^3^/mL TCID_50_ (2 mL/pig) through the intranasal (IN) route, and the challenged pigs were then maintained in their groups to observe the transmission of the virus to the unchallenged contact pigs. On the same day, all 3 pigs in group 2 and group 3 were injected with 0.32 mM (0.04 mM/kg body weight) and 1.6 mM (0.2 mM/kg body weight) DiNap prepared in 2 mL of DMSO behind the base of each ear intramuscularly (IM), respectively. All 3 pigs in group 1 constituted the nontreated challenged control (NC) group. The treatment with DiNap was repeated with the same concentrations at 3 dpc. After this, the pigs were observed daily for clinical signs up to 28 dpc. The pigs were weighed at 0 (before challenge) and 28 dpc. Serum samples were collected at 0 (before challenge), 7, 14, 21, and 28 dpc for virological and serological assays. At the end of 28 dpc, all pigs were euthanized, and the results of pathological examinations, including the presence of gross lung lesions, were recorded by an expert pathologist based on the standard scoring system, as described previously [57]. In addition, lung tissue samples were collected in 10% neutral buffered formalin for the examination of microscopic changes. Furthermore, additional lung tissue samples were collected and stored at −80 °C until analysis. Viral loads were quantified in the serum and lung tissue samples. The animal experiment protocols were approved by the Chonbuk National University Institutional Animal Care and Use Committee (approval number: 2012-0025).

#### 4.5.1. Serum ELISA

PRRSV-specific antibodies (IgG) were detected in the serum samples from each group by using a commercially available ELISA kit (IDEXX PRRS X3 Ab Test, IDEXX Laboratories, AG, Berne, Switzerland) following the instructions supplied by the manufacturer. An S/P ratio (ratio between the net optical density of the test samples and the net optical density of the positive control) ≥ 0.4 was considered positive for the PRRSV antibody.

#### 4.5.2. Lung Tissue Scoring and Pathology

At the end of 28 dpc, the lungs of the euthanized pigs were scored for gross lesions immediately after necropsy [57]. The microscopic lesions were also scored in lung tissue sections by staining with hematoxylin and eosin (H & E). The scoring of the microscopic lesions was performed according to a scale from 0 to 3 depending on the degree of inflammation (especially the presence of interstitial pneumonia), where 0 indicated no lesions, 1 indicated mild interstitial pneumonia, 2 indicated moderate/multifocal interstitial pneumonia, and 3 indicated severe intense/diffuse interstitial pneumonia.

#### 4.5.3. Quantification of Viral Loads in Serum and Lung Tissue

Viral RNA was extracted from the serum and lung tissue samples using the MagMAX™ Viral RNA Isolation Kit (Ambion, Applied Biosystems, Life Technologies, Inc., Carlsbad, CA, USA) and a total RNA extraction kit (Hybrid-RTM, GeneAll, Seoul, Korea), respectively, according to the manufacturer’s instructions. The viral loads in the serum and tissues were quantified by real-time reverse transcription-polymerase chain reaction (qRT-PCR) using a TaqMan^®^ assay based on a previous study [58]. The primer and probe sequences were as follows: forward primer TGTCAGATTCAGGGAGRATAAGTTAC; probe 6-FAM TGTGGAGTTYAGTYTGCC; and reverse primer ATCARGCGCACAGTRTGATGC. A one-step quantitative real-time RT-PCR (qRT-PCR) kit (AgPath-ID™ One-Step RT-PCR, Ambion, Applied Biosystems, Life Technologies, Inc., Carlsbad, CA, USA) was used to measure the viral loads in the serum and lung tissues. Polymerase chain reaction (PCR) amplification was performed with a 7500 Fast Real-time PCR system (Applied Biosystems, Foster City, CA, USA) according to the manufacturer’s guidelines. One-step qRT-PCR was performed in a total of 25 µL containing 5 µL of template RNA, 12.5 µL of 2× RT-PCR buffer, 0.5 µL of each forward and reverse primer (20 pmol, at a final concentration of 0.8 pmol), 0.2 µL of the TaqMan^®^ probes (25 pmol, at a final concentration of 1 pmol), 0.5 µL of RNAse inhibitor (40 U/µL; RiboLock^TM^, Thermo Fisher Scientific Inc., Germany), 1 µL of 25X RT-PCR enzyme mix, and 4.8 µL of nuclease-free water. The cycling conditions were as follows: (a) Reverse transcription for 10 min at 45 °C; (b) a 10-min activation step at 95 °C; and (c) 40 cycles of 15 s at 95 °C and 45 s at 60 °C. Samples with a threshold cycle (C_t_) of 35 cycles or fewer were considered positive. A standard curve was generated from known virus titers and used to calculate the amount of PRRSV in each sample by converting the C_t_ value to the TCID_50_/mL equivalent.

### 4.6. Data Analysis

Repeated measures analysis of variance (ANOVA) was used to analyze the significance of differences among the treatment groups for the in vitro evaluation of the effect of DiNap on PRRSV replication in cells and for the serum viremia and serum ELISA (anti-PRRSV IgG) results. The Mann–Whitney U test (a nonparametric t test) was used to analyze the significance of differences among groups for ADWG, lung lesions (gross and microscopic) and viral loads in lung tissues. *p* < 0.05 indicated a statistically significant difference. Graph Pad Prism 5.0.2 (GraphPad Software, Inc., CA, USA) was used to produce the graphs, while statistical analysis was performed using SPSS Advanced Statistics 17.0 software (SPSS, Inc., Chicago, IL, USA).

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
