# Peer review of "Evaluation of the Inhibitory Effects of (E)-1-(2-hydroxy-4,6-dimethoxyphenyl)-3-(naphthalen-1-yl)prop-2-en-1-one (DiNap), a Natural Product Analog, on the Replication of Type 2 PRRSV In Vitro and In Vivo"

_molecules, 2019, doi:10.3390/molecules24050887_

Reviewer 1 Report

This is a well written manuscript describing a thorough investigation of DiNAP effect on PRRSV. The experiments and conclusions were logically reported.  Two small notes: 1) On lie 52 the word "ago" is missing from the phrase "more than two decades ago"  2) It might be interesting to investigate a weight gain control of uninfected pigs vs uninfected pipgs dosed with DiNAP to determine if weight gain is due to virus relief or a different biological mechanism. 

Author Response

Reviewer 1:

This is a well written manuscript describing a thorough investigation of DiNAP effect on PRRSV. The experiments and conclusions were logically reported. 

Two small notes:

1) On line 52 the word "ago" is missing from the phrase "more than two decades ago" 

à This sentence has been corrected as suggested (line 58 in the revised manuscript).

2) It might be interesting to investigate a weight gain control of uninfected pigs vs uninfected pigs dosed with DiNAP to determine if weight gain is due to virus relief or a different biological mechanism. 

à A comparison of weight gain between uninfected pigs in the control group and uninfected pigs dosed with DiNap was impossible because noninfected pigs in the NC group became viremic at 7 dpc. However, we concluded that the significant increase in weight gain observed in the groups treated with DiNap was due to DiNap treatment because the weight gain in the treated groups increased in a dose-dependent manner (Figure 4).

Reviewer 2 Report

In this manuscript, Amina Khatun and colleagues studied DiNap, an analog of natural products (chalcone flavokawain), functions when against PRRSV infection in vitro and in vivo. The antiviral assays and MTT cytotoxicity assays were performed in MARC-145 and PAMs cell lines. The virus viremia and pathogenicity properties were also examined in animal experiments. The study showed that DiNap could inhibit VR2332 replication in both in vitro and in vivo. Additionally, virus transmission was also delayed approximately one week in uninfected contact pigs within the same group when treated with DiNap.

This is an interesting finding which could help us understand the potential drug that can be used to against PRRSV infection. However, during the review of this work, several questions have arisen that will require modifications, corrections or changes to the manuscript:

1.    Fig1. Authors only showed the DiNap antiviral effect against VR2332 at MOI 0.01. Please explain the rationale to choose 0.01 MOI of virus infection. Also, please add DiNap in vitro experiments against the various amount of virus infection from 0.01 to 1 MOI.

2.    Fig 6. The data of virus challenged pigs should separate with two other pigs when calculate the residual viral loads since there is a delayed transmission in group 2 and group 3. 

3.    Line 223 to 226, Authors claimed that “More importantly, a single injection of DiNap was able to delay virus transmission by approximately one week with direct contact between the infected (virus challenged) and uninfected (not virus challenged) pigs.” However, from Line 317 to 318, the authors described a repeated DiNap treatment for experimented pigs.

4.    Line 163 change ADWG to Average Daily Weight Gain (ADWG).

5.    Line 316 change (IM) to Intramuscular (IM).

6.    Describe experiment methods briefly in each figure at figure legend beside the “Materials and Methods” section.

Author Response

Reviewer 2:

In this manuscript, Amina Khatun and colleagues studied DiNap, an analog of natural products (chalcone flavokawain), functions when against PRRSV infection in vitro and in vivo. The antiviral assays and MTT cytotoxicity assays were performed in MARC-145 and PAMs cell lines. The virus viremia and pathogenicity properties were also examined in animal experiments. The study showed that DiNap could inhibit VR2332 replication in both in vitro and in vivo. Additionally, virus transmission was also delayed approximately one week in uninfected contact pigs within the same group when treated with DiNap.

This is an interesting finding which could help us understand the potential drug that can be used to against PRRSV infection. However, during the review of this work, several questions have arisen that will require modifications, corrections or changes to the manuscript:

1.    Fig1. Authors only showed the DiNap antiviral effect against VR2332 at MOI 0.01. Please explain the rationale to choose 0.01 MOI of virus infection. Also, please add DiNap in vitro experiments against the various amount of virus infection from 0.01 to 1 MOI.

à Because PRRSV (VR2332) efficiently replicates in both of these cell types, an MOI of 0.01 was used to evaluate the antiviral activity of various drugs against PRRSV in other previous studies too. In fact, VR2332 has been inoculated at 103 TCID50/ml for routine virus passage in our laboratory. Because an MOI of 0.01 is equal to 103 TCID50/ml, we believe that a reasonable amount of virus was used for antiviral evaluation.

2.    Fig 6. The data of virus challenged pigs should separate with two other pigs when calculate the residual viral loads since there is a delayed transmission in group 2 and group 3. 

à The circles indicate the pig challenged with virus in each group. The challenged pigs in the NC group showed significantly higher viral loads in the lungs than the challenged pigs in the groups treated with DiNap (Figure 6).

3.    Line 223 to 226, Authors claimed that “More importantly, a single injection of DiNap was able to delay virus transmission by approximately one week with direct contact between the infected (virus challenged) and uninfected (not virus challenged) pigs.” However, from Line 317 to 318, the authors described a repeated DiNap treatment for experimented pigs.

à The phrase “a single injection” was corrected to “two injections” (line 282 in the revised manuscript). We apologize for this mistake.

4.    Line 163 change ADWG to Average Daily Weight Gain (ADWG).

à The text has been corrected as suggested (the line 195 in the revised manuscript).

5.    Line 316 change (IM) to Intramuscular (IM).

à The text has been corrected as suggested (line 374 in the revised manuscript).

6.    Describe experiment methods briefly in each figure at figure legend beside the “Materials and Methods” section.

à The figure legends have been corrected as suggested.

Reviewer 3 Report

In the presented manuscript, Khatun et al report the generation of a potential anti PRRSV compound and its effect in vitro and in vivo. Whilst the topic is of general interest and very relevant to the field, there are some concerns especially regarding the data derived from the animal experiment and the testing and purity of the compound used. Therefore I would recommend a major revision. major points: - how was the chemical synthesis controlled and the amount of the final product quantified? Please put these results in the results section. At 32% purity, how can the authors be sure that DiNap is the pharmacologically active compound? - the number of animals in the experimental setup is too low for statistical analysis and the sampling interval is very long. Also, no clinical data are provided for the course of the experiment (fever / clinical symptoms etc…) minor points: - please check the language of your manuscript, especially regarding grammar - line 49-51: to my knowledge, PRRSV 1 and 2 are by now recognised as 2 different species - Fig. 1: please replace ‘relative cells viability’ by relative cell viability - whilst the effect in the MTT essay was not statistically significant, one still has to take the biological relevance of 20% of the cells dying after 48h at higher concentrations into consideration - line 115 and 124: 10^4.18 and 10^4.08 are not close to being 100-fold lower than 10^5.55. Please be specific. This also applies to later comparisons, such as in line 184 and following - Figure 2: I am a bit surprised by the coincidence of peak titers at day 7 pc in the NC. The sampling interval seems too long. If you don’t have more samples available, please provide evidence from the literature about average times from infection to virus shedding and infection to viremia to increase confidence in the data. Also, the peak in group 3 is not lower than in group 2, just delayed. - are any data available about bioavailability of the drug and adverse effects at the injection site? How was the drug injected? - it would be very interesting to have a shorter interval of weighing - please replace PRRS-free animals by PRRSV free animals and explain how this status was confirmed. - wouldn’t it be more biologically relevant to test for virus secretion than for IgG response? - please rewrite the discussion. In its current form it reads like a summary of the results section.

Author Response

Reviewer 3:

In the presented manuscript, Khatun et al report the generation of a potential anti PRRSV compound and its effect in vitro and in vivo. Whilst the topic is of general interest and very relevant to the field, there are some concerns especially regarding the data derived from the animal experiment and the testing and purity of the compound used. Therefore I would recommend a major revision.

major points:

- How was the chemical synthesis controlled and the amount of the final product quantified? Please put these results in the results section. At 32% purity, how can the authors be sure that DiNap is the pharmacologically active compound?

à The phrase 32% yield in the manuscript means that we obtained the compound from the starting materials with a yield of 32%, not a purity of 32%. As we mentioned in the methods section, we purified the compounds by MPLC with up to 95% purity. The purity of the compound was confirmed by analytical HPLC, along with 1H NMR and 13C NMR. To clarify the purification of the compounds, we also added the following sentence on Page 11, lines 306-308: “The purity of all tested compounds was confirmed to be higher than 95% by HPLC analysis performed with a dual-pump Shimadzu LC-6AD system equipped with a VP-ODS C18 column (4.6 mm×250 mm, 5 mm, Shimadzu)”.

– The number of animals in the experimental setup is too low for statistical analysis and the sampling interval is very long. Also, no clinical data are provided for the course of the experiment (fever / clinical symptoms etc…)

à Because it took approximately 3 months to synthesize 6 mg of DiNap at the laboratory scale, only 6 pigs in two groups could be treated with the limited amount of DiNap: a total of 0.32 mg (0.64 mM/pig) or 1.6 mg (3.2 mM/pig) DiNap was used for each piglet weighing approximately 8 kg.

In addition, as shown in Figures 2 and 3, because the pigs in the treated groups remained uninfected until 7 dpc, we decided to continue the animal experiment longer than originally planned. Since weight gain and lung lesions are the most important and obvious parameters that are directly related to farm productivity, those parameters are generally used for clinical data for PRRSV infection, rather than fever or clinical symptoms. In addition, PRRSV (VR2332) infection could be precisely detected and quantified based on the levels of viremia and PRRSV-specific antibody production. However, based on previous studies and our own experience, VR2332 exhibits a moderate level of virulence and does not induce obvious fever or clinical signs in pigs.

minor points:

- please check the language of your manuscript, especially regarding grammar

à The revised manuscript was re-edited for English by a native speaker.

- line 49-51: to my knowledge, PRRSV 1 and 2 are by now recognised as 2 different species

à To our knowledge, PRRSV 1 and 2 are classified as 2 genotypes. Please let us know if the reviewer has different information.

- Fig. 1: please replace ‘relative cells viability’ by relative cell viability

à The text has been corrected as suggested

- whilst the effect in the MTT essay was not statistically significant, one still has to take the biological relevance of 20% of the cells dying after 48h at higher concentrations into consideration

à A relevant comment was added to the discussion in lines 273-275. “Even though approximately 20% cell death was observed when DiNap was added at 0.06 mM, no significant side effects or toxicity were observed after treatment in pigs.”

- line 115 and 124: 10^4.18 and 10^4.08 are not close to being 100-fold lower than 10^5.55. Please be specific. This also applies to later comparisons, such as in line 184 and following

à This passage was corrected as suggested in lines 128-138 in the revised manuscript. We apologize for this mistake.

- Figure 2: I am a bit surprised by the coincidence of peak titers at day 7 pc in the NC. The sampling interval seems too long. If you don’t have more samples available, please provide evidence from the literature about average times from infection to virus shedding and infection to viremia to increase confidence in the data. Also, the peak in group 3 is not lower than in group 2, just delayed. - are any data available about bioavailability of the drug and adverse effects at the injection site? How was the drug injected?

à Based on other studies and our own experience, viremia is generally detected as early as 3 days after infection with various PRRSV strains, peaks at 7-10 days after infection and disappears approximately 28 days after infection, depending on the age of the pigs (Figure 1). The viremia trends were very similar regardless of which PRRSV strains were used for challenge (Figure 2).

Figure   1. Viremia trends after PRRSV infection [Sun D et al, Attempts to enhance   cross-protection against porcine reproductive and respiratory syndrome   viruses using chimeric viruses containing structural genes from two   antigenically distinct strains [1].

Figure   2. Viremia trends after PRRSV infection with various PRRSV strains (unpublished   data)

 Because DiNap was injected intramuscularly and the virus was challenged intranasally, we do not think the injected drug directly affected virus replication. We speculate that the effect of DiNap treatment was maintained for approximately 1-2 weeks after inoculation and would like to evaluate the duration of DiNap treatment in more detail with multiple treatments up to 28 days after challenge in the future.

References:

1. Sun D, Khatun A, Kim WI, Cooper V, Cho YI, Wang C, Choi EJ, Yoon KJ (2016) Attempts to enhance cross-protection against porcine reproductive and respiratory syndrome viruses using chimeric viruses containing structural genes from two antigenically distinct strains. Vaccine 34 (36):4335-4342. doi:10.1016/j.vaccine.2016.06.069

- it would be very interesting to have a shorter interval of weighing

à We are very sorry that we did not measure the weight every week during the 28 days of the animal experiment. Because PRRSV infection could be precisely detected and quantified based on the levels of viremia and PRRSV-specific antibody production, we thought that weight gains between 0 and 28 dpc were sufficient to evaluate the effects of DiNap on weight gain.

- please replace PRRS-free animals by PRRSV free animals and explain how this status was confirmed.

à This phrase was corrected as suggested in the Materials and Methods section (line 371).

The pigs were purchased from a PRRSV-free commercial farm for the animal experiment. Then, the authors confirmed that all pigs were negative for PRRSV by virological and serological tests using qRT-PCR (TaqMan) and ELISA, respectively, which was described clearly in “Animal Experiment and Samples” in the Materials and Methods section of the manuscript between lines 373 and 374.

- wouldn’t it be more biologically relevant to test for virus secretion than for IgG response?

à Viremia is the most obvious direct evidence of PRRSV infection, and the PRRSV-specific IgG response is indicative of PRRSV infection.

- please rewrite the discussion. In its current form it reads like a summary of the results section.

à The discussion was revised as suggested. Since this is the first study to confirm the antiviral effects of DiNap against PRRSV infection in pigs, there is no previous study to compare with the current study. Please take this into consideration.

Round  2

Reviewer 3 Report

We thank the authors for their comments and clarifications.

Given the comments on substance purity, you state in  the reply:

As we mentioned in the methods section, we purified the compounds by MPLC with up to 95% purity. 

I am not an expert in pharmacological substance purification, as will be most of the potential readers of your manuscript, so please add this statement like this in the methods section and also explain why your purify with up to 95% purity by MPLC and then measure a purity larger than 95% by HPLC. Making this understandable to the none expert reader is important to generate trust in your experiments. 

Please mention in the text of the manuscript the limited availability of your substance and therefore the limited scale of the animal experiment. Also, please clearly state that this makes the results obtained by this experiment preliminary. As additional data on the animal experiment can obviously not be provided anymore, I would kindly ask you to consider checking the weight gain at least once per week, ideally daily, and to do so with clinical signs as well. Even if you know your infection model well (how do you explain the coincidence of peak titers in your NC group if only 1 animal in the group was inoculated with the virus?), you are now introducing a pharmacological substance, which might also have effects on the animals that you want to monitor closely. Therefore, closer monitoring of virus titers is important (at least every 3 days), as is the monitoring of virus secretion (which can easily be monitored daily with nasal swabs), especially if you want to discuss reduced transmission (effect of substance in infected pig vs pig to be infected). 

This also holds true for monitoring of the injection site, as this is important evidence of how well the infection was tolerated. Ideally, the injection site should be monitored postmortem histologically. 

Please mention the injection site and volume of injection in your materials and methods section! And also why you chose to use this way of application. 

Please have a look at the ICTV homepage regarding PRRSV taxonomy.

https://talk.ictvonline.org/ictv/proposals/2015.014a-cS.A.v3.Arteriviridae_sprev.pdf

Author Response

Given the comments on substance purity, you state in the reply:

As we mentioned in the methods section, we purified the compounds by MPLC with up to 95% purity. 

I am not an expert in pharmacological substance purification, as will be most of the potential readers of your manuscript, so please add this statement like this in the methods section and also explain why your purify with up to 95% purity by MPLC and then measure a purity larger than 95% by HPLC. Making this understandable to the none expert reader is important to generate trust in your experiments. 

à “We purified the compounds by MPLC up to 95% purity” in the rebuttal means that we purified the compounds by the preparative MPLC and the purity of the compounds was confirmed to be higher than 95% by the analytical HPLC.

To clarify the statement in the manuscript, we changed the sentences in the manuscript as follows.

“The final products were purified by preparative medium-pressure liquid chromatography (MPLC) (Biotage Isolera One instrument) on a silica column (Biotage SNAP HP-Sil)” in page 11, lines 303-304

The purity of all tested compounds was confirmed to be higher than 95% by analytical HPLC analysis performed with a dual pump Shimadzu LC-6AD system equipped with VP-ODS C18 column (4.6 mm×250 mm, 5 mm, Shimadzu)” in Page 11, lines 309-311

Please mention in the text of the manuscript the limited availability of your substance and therefore the limited scale of the animal experiment. Also, please clearly state that this makes the results obtained by this experiment preliminary.

à This information was added to the discussion in the revised manuscript as suggested. Thank you for your suggestion (Page 11, lines 288-291)

As additional data on the animal experiment can obviously not be provided anymore, I would kindly ask you to consider checking the weight gain at least once per week, ideally daily, and to do so with clinical signs as well. Even if you know your infection model well (how do you explain the coincidence of peak titers in your NC group if only 1 animal in the group was inoculated with the virus?), you are now introducing a pharmacological substance, which might also have effects on the animals that you want to monitor closely. Therefore, closer monitoring of virus titers is important (at least every 3 days), as is the monitoring of virus secretion (which can easily be monitored daily with nasal swabs), especially if you want to discuss reduced transmission (effect of substance in infected pig vs pig to be infected). This also holds true for monitoring of the injection site, as this is important evidence of how well the infection was tolerated. Ideally, the injection site should be monitored postmortem histologically. 

à First of all, we are sincerely sorry that we didn’t collect the blood samples and measure the weight gains more often. We agree that we could have much better results if we did the sample collections more often. In the next experiment, we will increase the number of samples collection and add nasal swab sampling and pathology on injection sites as well to reflect your suggestions. Thank you for your valuable comments.

As the reviewer indicated, similar peak titers observed in the NC group even though only #1 pig in the group was inoculated with the virus. Because those 3 pigs were housed in the same room and shared feed and water together, we think the virus transmission from the challenged pigs to the other 2 pigs happened in 1-3 days after virus challenge by nasal transmission as virus challenge was conducted intranasally (probably, pig no 3 was infected earlier than pig no 2). The reason for this assumption is that viremia titers of piglets infected with virus transmission (pigs no 2 and 3) were observed at lower levels at 7 dpc and the slope of viremia levels until 28 dpc (pig no 2) was smaller as compared with the challenged piglets (pig no 1).

Please mention the injection site and volume of injection in your materials and methods section! And also why you chose to use this way of application. 

à The information was added as “On the same day, all 3 pigs in group 2 and group 3 were injected with 0.32 mM (0.04 mM/kg body weight) and 1.6 mM (0.2 mM/kg body weight) DiNap prepared in 2 ml of DMSO behind the base of each ear intramuscularly (IM), respectively.” (Page 13, lines 381-383). Behind the base of each ear is the most popular site for vaccination and other treatments for pigs.

Please have a look at the ICTV homepage regarding PRRSV taxonomy.

https://talk.ictvonline.org/ictv/proposals/2015.014a-cS.A.v3.Arteriviridae_sprev.pdf

--> The information has been corrected as suggested (Page 2, lines 55-58). Thank you for the information.